# Seasonal and Spatial Variations of Dissolved Organic Matter Biodegradation along the Aquatic Continuum in the Southern Taiga Bog Complex, Western Siberia

Tatiana V. Raudina [1,2,*], Sergei V. Smirnov [1,2], Inna V. Lushchaeva [2], Georgyi I. Istigechev [2], Sergey P. Kulizhskiy [2], Evgeniya A. Golovatskaya [1], Liudmila S. Shirokova [3,4] and Oleg S. Pokrovsky [3]

[1] Institute of Monitoring of Climatic and Ecological Systems, Siberian Branch of the Russian Academy of Sciences, 634055 Tomsk, Russia
[2] BIO-GEO-CLIM Laboratory, Tomsk State University, 634050 Tomsk, Russia
[3] Geoscience and Environment Toulouse, UMR 5563 CNRS, Avenue Edouard Belin, 31400 Toulouse, France
[4] N. Laverov Federal Center for Integrated Arctic Research, UrB Russian Academy of Science, 163020 Arkhangelsk, Russia
* Correspondence: tanya_raud@mail.ru

**Abstract:** The inland aquatic ecosystems play a significant role in the global carbon cycle, owing to the metabolism of terrestrially derived organic matter as it moves through fluvial networks along the water continuum. During this transport, dissolved organic matter (DOM) is microbial processed and released into the atmosphere, but the degree and intensity of this processing vary greatly both spatially and temporally. The Western Siberian Lowlands is of particular interest for a quantitative assessment of DOM biodegradation potential because the global areal-scale effects of DOM biodegradation in abundant surface organic-rich waters might be the highest in this region. To this end, we collected water samples along a typical aquatic continuum of the Bakchar Bog (the north-eastern part of the Great Vasyugan Mire) and, following standardized protocol, conducted an experimental study aimed at characterizing the seasonal and spatial variability of dissolved organic carbon (DOC) biodegradability. The biodegradable DOC fraction (BDOC) over the exposure incubation period ranged from 2% to 25%. The natural aquatic continuum "mire–forest–stream–river" demonstrated the systematic evolution of biodegradable DOC among the sites and across the seasons. The highest biodegradation rates were measured during spring flood in May and decreased along the continuum. The maximum possible $CO_2$ production from DOM yielded the maximum possible flux in the range of 0.1 and 0.2 g C-$CO_2$ m$^{-2}$ day$^{-1}$ d, which is an order of magnitude lower than the actual net $CO_2$ emissions from the inland waters of the WSL. This study suggests that although the biodegradation of the humic waters of the WSL can sizably modify the concentration and nature of the DOM along the aquatic continuum, it plays only a subordinary role in overall C emissions from the lakes and rivers of the region.

**Keywords:** organic carbon; biodegradation; incubation; forest; river; mire; stream; soil water; peatland

## 1. Introduction

Dissolved organic matter (DOM) is one of the largest biologically available sources of the carbon pool in aquatic environments, and its dynamics are critical for local and global carbon cycles. During lateral transport along the aquatic continuum, i.e., from feeding sources, through headwater streams to large terminal water bodies, the modification or removal of dissolved organic matter occurs due to biotic and abiotic processes [1–5]. All microbial uptake mechanisms require an aqueous environment, since the soluble state seems to be a prerequisite for the diffusion of substrates across microbial cell membranes. Therefore, one of the key regulators of ecosystem metabolism in general and the rate of

carbon release to the atmosphere in particular is the biochemical transformation (mineralization) of DOM by bacteria leading to net $CO_2$ emission from soil and aquatic settings [6–8]. Bioavailability describes the potential of microorganisms to interact with DOM. The magnitude of this process is determined primarily by the sources of DOM and varies spatially and temporally. The dissolved organic matter characteristics that generally enhance its biodegradability are high contents of carbohydrates, organic acids and proteins, for which the hydrophilic fraction seems to be a good estimate. Numerous laboratory incubation studies report that between 10 and 40% of DOM in aquatic ecosystems may be subject to microbial degradation (see [9] for a review). The majority of these studies were focused on the quantification of bioavailable DOM (BDOM) in the river and lake waters of high latitude regions (e.g., [2,10–16]). There are numerous studies of DOM biodegradability performed either in Alaska and Canada [7,8,17–20], or in Eastern Siberia [6,14,21–24] and usually dealing with streams draining shallow peat, mineral or yedoma organic-rich soils. There are also several studies on the transformation of DOM along headwater catchments and mire complexes in Fennoscandia [25–29]. In contrast, there are only a few inland waters studies in the Western Siberian Lowlands (WSL) that quantify the magnitude and controlling factors of the DOM biodegradation potential in the peatlands of discontinuous and continuous permafrost [15,30]. Some studies demonstrated that aquatic and soil BDOC losses are significantly lower in regions without permafrost than in regions with discontinuous or continuous permafrost [2,8,31–34], but on the other hand, there are also a number of contrary cases [15,21,35,36]. Furthermore, overwhelmingly the biodegradation studies were conducted in mid-summer and did not describe seasonal patterns of biodegradable DOC. At the same time, relative biodegradability was shown to be highest during winter and spring floods in the Yukon, Alaska (35–53% BDOC) and Kolyma, Eastern Siberia (about 20% BDOC) rivers.

Thus, the available data demonstrate that there are still large uncertainties related to conflicting patterns of DOC biodegradability with respect to seasonality and permafrost extent, with only a few data available from the WSL. This environmentally important and highly vulnerable region exhibits intense paludification (in some parts up to 70–80%), which leads to a predominance of organic-rich inland waters. Therefore, the WSL is of particular interest for a quantitative assessment of DOM biodegradation potential because the global areal-scale effects of DOM biodegradation in abundant surface waters might be the highest in this region. In particular, Karlsson et al. [37] demonstrated the sizable emission of $CO_2$ from the inland waters of Western Siberia, with net emission flux (0.1 Pg C $y^{-1}$) being comparable (30 to 50%) to that of land primary production. It is thus possible that, among various drivers, the degradation of allochthonous DOM in the water column of stagnant and flowing surface waters might contribute to this elevated $CO_2$ flux; however, experimental assessments of these processes were quite limited.

In this work, we focused on a small wetland watershed located within the Great Vasyugan Mire (GVM), a natural UNESCO heritage site of world value exhibiting unique location (between the southern taiga and forest-steppe biomes) and size (67,800 $km^2$, where water surfaces occupy as much as 30%). By its sheer size, the mire controls globally relevant carbon stocks and GHG emissions and removals, and regulates regionally important fluxes of dissolved organic matter and trace metals [37–41]. The spatial and temporal variation in water chemistry within this mire is relatively well studied [42–47], but the susceptibility of this mire's DOM to biodegradation remains completely unknown. Here, we conducted an experimental study aimed at characterizing the seasonal and spatial variability of the DOC biodegradability of water samples collected along a typical aquatic continuum. Since DOC concentrations are known to exhibit significant spatial and temporal variation, and small streams draining peatlands exert a critical influence on the hydrochemistry of the whole fluvial network, we chose the catchment area of the Klyuch River ($S_{watershed}$ = 76 $km^2$), which is the tributary of the second-order Bakchar River ($S_{watershed}$ = 2040 $km^2$) located entirely within the GVM. Through incubation experiments on aerobic biodegradation, we monitored the resulting changes in the amount and quality of DOM. We hypothesized

the preferential biodegradation of headwater streams and soil pore waters relative to low reaches of the large river, in accordance with general settings of aquatic continuums. We further expected higher biodegradation during the summer and autumn periods compared to the spring flood, after snowmelt due to the higher concentration of autochthonous, more bioavailable DOM during the baseflow [9]. Testing these hypotheses constituted the first objective of this study. The second objective was quantifying possible atmospheric $CO_2$ flux related to the DOM biodegradation of the studied water objects, and comparing this flux with global values of $CO_2$ evasion from surface waters in permafrost-free parts of the WSL as estimated in a recent work of Karlsson et al. [37].

## 2. Materials and Methods

### 2.1. Study Area and Field Smpling

The study area is located in the southeastern part of the West Siberian Lowland, on the Ob–Irtysh interfluve within the Vasyugan plain (Figure 1). The territory is a hilly waterlogged plain with a decrease in absolute elevations in the northern and northeastern direction from 160 to 100 m. Significant mire areas are distinguished by surfaces with fluvioglacial and denudation-accumulative relief. For the floodplains and above-floodplain terraces of the Bakchar River, the typical relief type is erosion-accumulative. The dominant lithology is fluviolacustrine loams and clays with a thickness of 40–60 m on the interfluve of the Bakchar and Iksa rivers. The mire deposits are represented by peat, the accumulation of which began about 8–9 thousand years ago. There are no deep-seated relict permafrost occurrences and cryogenic processes are seasonal. This territory belongs to the southern taiga subzone and it is characterized by excessive moisture and insufficient heat supply. The climate is continental, with mean annual temperature ranging from $-0.91\,°C$ to $+1.6\,°C$. The annual precipitation ranges from 450 mm to 500 mm with maximum values during the warm period (July–August) of the year, which exceeds evapotranspiration (300–360 mm) (according to Bakchar weather station, http://meteo.ru/, accessed on 20 September 2022).

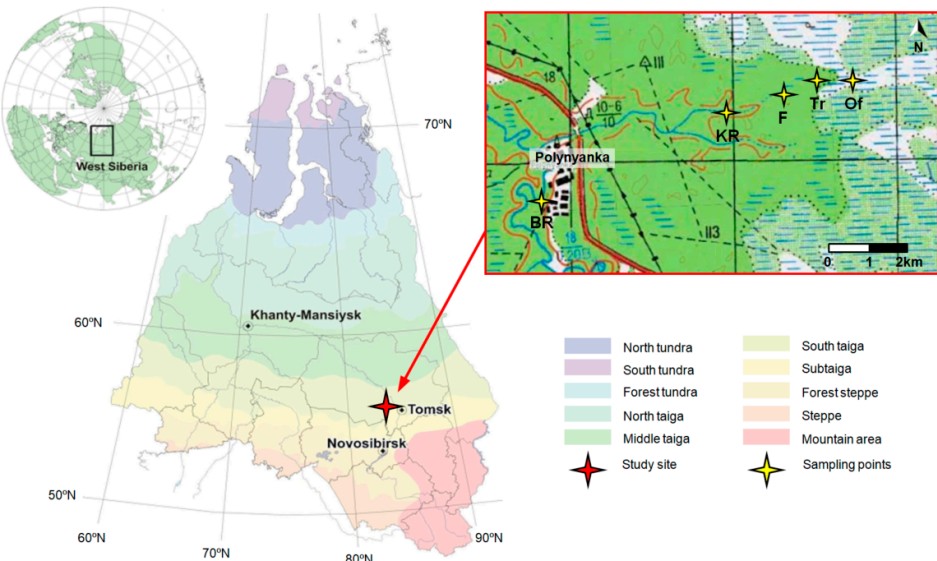

**Figure 1.** Map of the study site in the Western Siberian Lowland with study site (red asterisk) and sampling points along the aquatic continuum as shown by yellow asterisks: Of—open sedge-sphagnum fen; Tr—tall ryam (pine–shrub–sphagnum phytocenosis with high pine trees); F—waterlogged pine-birch forest; KR—Klyuch River (small mire stream); BR—Bakchar River.

The territory of the northeastern part of the Great Vasyugan Mire, where our research was carried out, is called the Bakchar Bog (near Polynyanka village, 56°57′17.10′′ N; 82°30′32.37′′ E) and corresponds to the catchment area of the Klyuch River, having a watershed size of 76 km² (Figure 1). This river is the right tributary of the Bakchar River

($S_{watershed}$ = 2040 km$^2$), formed by the confluence of two mire streams that collect runoff from the northern and eastern parts of oligotrophic raised mire. The runoff formation in the river is determined by the water saturation of peat and its properties. In the winter, the river freezes completely; in the spring, the runoff in the river appears only after the water levels rise due to snowmelt at the mire surfaces; in the summer-autumn season, the river is fed essentially by mire waters. According to Savichev et al. [48], the area of the Klyuch River catchment is 77% waterlogged, and peat thickness reaches 4 m. The upper part of the mire peat profile, with a thickness of about 1–1.5 m, consists mainly of fibric peat, while the deeper layers are composed mainly of hemic or sapric peat.

An important feature of the studied site is the regular micro-landscapes change in the direction from the river to the central part of the mire (Figure 2). The landscape along the chosen aquatic continuum is represented by waterlogged mixed forest, tall ryam (pine–shrub–sphagnum phytocenosis with high pine), low ryam (pine–shrub–sphagnum phytocenosis with low pine) and ridge–hollow or ridge–pool complexes. In the latter, the spatial heterogeneity of the microtopography is dominated by moss hummocks/mounds, tussocks, hollows, fens and small depressions. The vegetation types covering this site are typical for the mires of the southern taiga zone of Western Siberia. The vegetation of the forest is dominated by *Pinus sibirica*, *Betula pubescens*, *Picea obovata*, *Rosa acicularis*, *Ledum palustre*, *Carex cespitosa*, *Calla palustris*, *Menyantes trifoliata* and *Sphagnum angustifolium*, while the pine–shrub–sphagnum phytocenosis are covered by *Pinus silvestris*, *Chamaedaphne calyculata*, *Ledum palustre* and *Sphagnum fuscum*. On the open fens and hollows there are *Chamaedaphne calyculata*, *Andromeda polifolia*, *Oxycoccus microcarpus*, *Eriophorum vaginatum*, *Carex rostrata* and mosses (*S. balticumand*, *S. angustifolium*, *S. magellanicum* and *S. fuscum*).

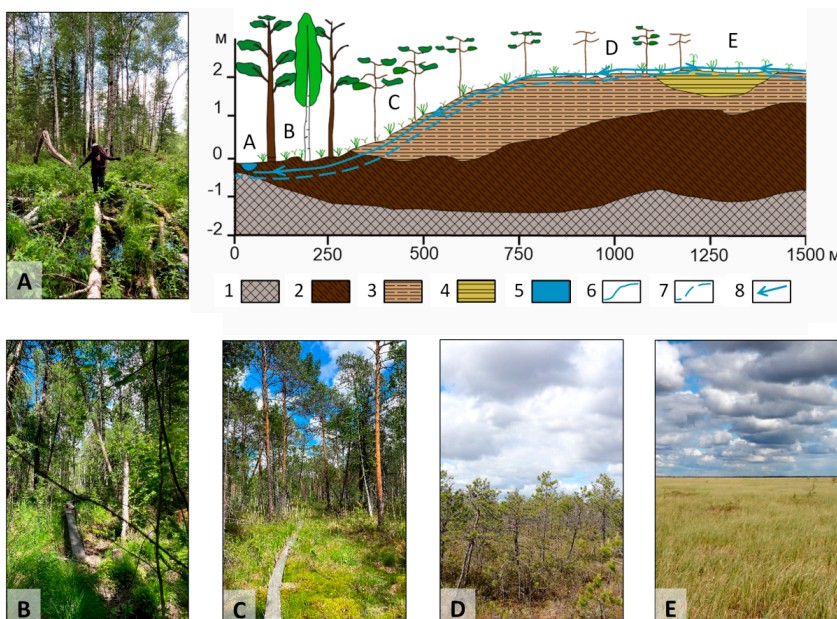

**Figure 2.** The aquatic continuum of the study site represented by Klyuch River (**A**), waterlogged mixed forest (**B**), tall ryam (pine−shrub−sphagnum phytocenosis with high pine trees) (**C**); low ryam (pine−shrub−sphagnum phytocenosis with low pine trees) (**D**); open sedge−sphagnum fen (**E**). The numbers on the diagram represent the following: 1—sedge or sedge−wood mainly highly decomposed peat (sapric); 2—wood−grass and wood−sphagnum medium decomposed peat (hemic); 3—sphagnum (dominated by *S. magellanicum*, and *S. fuscum*), grass−sphagnum, pine−cotton and pine shrub mostly undecomposed and medium decomposed peat (fibric and hemic); 4—sphagnum and sedge−sphagnum undecomposed peat (fibric); 5—mire stream (Klyuch River); 6—water level (20.05.2015/18); 7—water level (20.08.2015/18); 8—water flow in May (surface streamflow is absent during summer-autumn drought period).

We chose three locations belonging to mire landscape profile and two points in the Bakchar and Klyuch Rivers (Table S1). The Bakchar River was sampled near its confluence with the Klyuch River. These locations correspond to an aquatic continuum in the direction of hydrological flow, from stagnant surface and pore waters of pine–shrub–sphagnum, to the sedge–sphagnum ombrotrophic mires, to the flowing streams and a larger river. Water samples were collected in pre-cleaned PVC jars and kept refrigerated until their arrival at the laboratory, which occurred within 2–3 h of collection. For rivers, water was sampled by immersing a bottle 20–30 cm below the surface. In other cases, water was taken at depth by digging a large pit ($50 \times 50$ cm area, 50 cm depth) and allowing the surrounding gravitational water to fill it up to the depth of 10–20 cm. The sampling took place during three field campaigns in 2020: in the spring period on May 20th; at the end of summer on August 3rd; late autumn on October 15th at the beginning of first frosts. At each sampling point, the pH, water temperature, specific conductivity (Cond, $\mu S\ cm^{-1}$) and dissolved oxygen were measured in the field using a multiparameter instrument (MULTI 3430 SET, WTW, Germany).

*2.2. Experimental Setup and Data Treatment*

After collection, all water samples were processed within 12 h according to the standardized dissolved organic carbon (DOC) incubation protocol presented in Vonk et al. [2]. Our previous tests conducted in this and other similar sites of DOM-rich humic waters of the WSL mires demonstrated that the DOC level and its optical properties remain stable over at least 12 h regardless of whether the water is filtered on-site, immediately after sampling, or upon arrival at the laboratory under refrigerated conditions. The whole procedure was carried out under sterile environmental conditions of a laminar hood box A100, created using an ultraviolet (UV) light, autoclaving and pre-combusting. In brief, initial samples were filtered through pre-combusted (450 °C during 4.5 h) glass-fiber filters (GF/F, Whatman, Maidstone, UK, 0.7 mm, diameter 47 mm) using sterile filtration units (Millipore, Burlington, MA, USA, 250 mL), and then 30mL of filtrate was poured into triplicate sets of 40 mL pre-combusted (550 °C during 4.5 h) dark borosilicate glass vials in triplicates. The vials were left in an incubator at 20 °C for 28 days in the dark to eliminate autotrophic respiration and photodegradation. Aerobic conditions were ensured with a sterilized cellulose stopper and gentle daily shaking. At each time point (T = 0, 2, 7, 14, 22, 28 days) the incubated samples (three replicates) were re-filtered through pre-combusted 0.7 µm GF/F filters using a sterilized Sartorius filter holder (25 mm diameter). Control runs were 0.22 µm sterile-filtered water, which was incubated in parallel to the experiments and re-filtered through 0.7 µm GF/F filters on the sampling day.

Part of the collected filtrate was acidified to pH 2 with 30 µL of concentrated HCl, tightly capped and then stored in the refrigerator pending DOC analysis. The stability of DOC concentration in filtered and acidified waters kept in the refrigerator has been confirmed by our previous tests in DOC-rich peatland waters [30]. Dissolved organic matter was measured by a high-temperature thermic oxidation method using a Shimadzu TOC-LCPN analyzer, with an uncertainty of 2%. The non-acidified portion of the filtrate was used for pH (uncertainty of $\pm 0.01$ pH units), specific conductivity ($\pm 0.1\ \mu S\ cm^{-1}$) and absorbance (250 to 800 nm, 1 nm step; Varicen, Cary 50 Scan. UV-Visible). In addition, we measured the initial total bacterial cell (TBC) using microscopic count with acridine orange and luminescent microscope Zeiss Axio Imager Z2 (Jena, Germany) according to the standard method [49]. The typical uncertainty of these measurements ranged from 10 to 20%. Control experiments did not demonstrate the presence of any countable cells in the observation fields.

Biodegradable DOC was calculated as the difference in DOC concentration before and after 28 days of incubation following the equation proposed by Vonk et al. [2]. All BDOC incubations were run in triplicate and the mean DOC loss was used to derive the percent loss from the initial DOC concentration, defined as BDOC (%). The apparent removal rate of DOC was calculated as the slope of linear regression at 28 days of incubations ($R_{DOC28}$).

Data from UV-visible absorbance measurements were used to calculate various indicators of DOC quality. The spectral slope (S) of absorbance spectra was determined by applying log linear fits across the wavelengths 275–295 nm ($S_{275-295}$) and 350–400 nm ($S_{350-400}$), and the spectral slope ratio ($S_R$) calculated as the ratio between the two [50]. The specific UV absorbance ($SUVA_{254}$ or absorbance at 254 nm normalized for DOC concentration, L mg$^{-1}$ m$^{-1}$), $S_R$, E2:E4 (absorbance at 254 and 436 nm) ratio, E2:E3 (absorbance at 250 and 365 nm) ratio and weight average molecular weight index (WAMW) were used to approximate for carbon aromaticity, molecular weight and DOM source [6,50–53].

Statistical processing of the data included the use of non-parametric Kruskal–Wallis H test and pair Wilcoxon–Mann–Whitney U tests ($p < 0.05$). The Kruskal–Wallis test was used for evaluation of the difference of each component among several simultaneous samplings. In the case of significant differences, a comparison of physico-chemical parameters and UV-visible absorbance indicators between water sampled in one main pair of seasons of each of the 5 sampling points was conducted using a non-parametric pair Wilcoxon–Mann–Whitney test. All calculations and graphics were performed using MS Excel 2010 and standard package of STATISTICA-12.

## 3. Results and Discussion

### 3.1. Initial Water Conditions across the Seasons

The mire and river waters differ in physico-chemical, optical parameters and DOC concentrations both seasonally and spatially (Table S2). The pH values ranged from 3.7 (open fen) to 8.5 (Bakchar River), with the highest values observed in autumn and the lowest in summer. No obvious patterns in the soluble salt concentrations (reflected by specific conductivity ≤49 μS cm$^{-1}$) in the mire waters by seasons and micro-landscapes were found. In the river waters, an increase in the specific conductivity (from 334 to 580 μS cm$^{-1}$) was observed in the summer-autumn low water period due to low water levels and slow water exchange (Figure 3). On the contrary, in the spring, with an increase in water level, temperature and oxygen saturation, a decrease in specific conductivity and pH occurred.

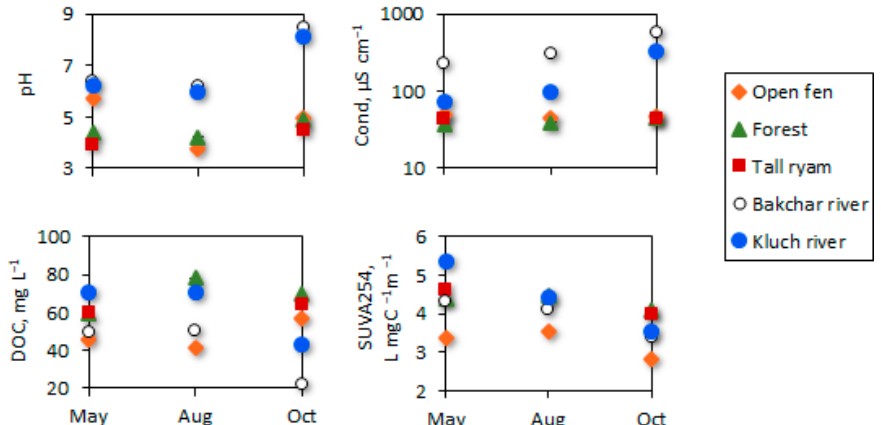

**Figure 3.** Some initial water parameters in different mire micro-landscapes and rivers by season.

The DOC concentrations within the mire micro-landscapes ranged from 42 to 80 mg L$^{-1}$ and reached a maximum in the August-October period (mean ± sd = 62 ± 14 mg L$^{-1}$). Riverine DOC concentrations (22–71 mg L$^{-1}$) usually peaked in spring (mean ± sd = 60 ± 15 mg L$^{-1}$) and decreased towards autumn (mean ± sd = 33 ± 15 mg L$^{-1}$). The total bacterial cell number in the studied water varied from $4.4 \times 10^6$ cells mL$^{-1}$ to $19 \times 10^7$ cells mL$^{-1}$, prevailing in autumn and changing in the order forest > fen > ryam > Kluch River > Bakchar River; the same spatial pattern was typical for DOC concentrations, $R^2 = 0.58$. The specific UV absorbency ($SUVA_{254}$, L mg$^{-1}$ m$^{-1}$) can act as a proxy for aromatic C, molecular weight and the source of DOM [52,54]. In the studied waters, the

$SUVA_{254}$ ranged between 2.8 and 5.3 L mg$^{-1}$ m$^{-1}$ over the study period. Spring conditions were characterized by the highest $SUVA_{254}$ values (3.4–5.3 L mg$^{-1}$ m$^{-1}$), whereas the lowest $SUVA_{254}$ values were observed in the autumn water samples (2.8–3.6 L mg$^{-1}$ m$^{-1}$). The normalized molecular weight (WAMW) values showed a similar seasonal pattern to $SUVA_{254}$ (r = 0.98, $p < 0.05$) for all the water samples from mire micro-landscapes and rivers. The water samples of the fen and the Bakchar River had a lower molecular weight and less aromatic DOM compared to other water samples ($SUVA_{254} \leq 4.3$ L mg$^{-1}$ m$^{-1}$). It has been shown that the spectral slope ratio ($S_R$) correlates with DOM molecular weight and source with an increasing ratio characterizing the decrease in molecular weight and a shift from DOC-rich black waters to optically clearer waters [50,55,56]. During the study period, $S_R$ fluctuated from 0.64 to 0.97. The lowest values observed in May indicated the input of higher molecular weight, aromatic terrestrial material, and then increased over the remainder of the study period showing a shift towards lower molecular weight and less aromatic DOM. Additional study of the E2:E3 and E2:E4 ratios found inverse linear relationships with $SUVA_{254}$ (r = −0.84 and −0.78, respectively, at $p < 0.001$) are consistent with the patterns described above. These results are therefore indicative of DOM concentration and composition variations throughout the study period, which were more pronounced in river waters compared to the stagnant surface and sub-surface waters of fen and forest. Presumably, increasing spring surface runoff leads to a rapid delivery to the river of allochthonous, highly aromatic organic matter originated from the peat leaching of the surface peat horizons at the upper parts of the aquatic continuum.

### 3.2. Variations in DOC Biodegradability

The BDOC values mirrored DOC concentration during incubation, varying spatially and seasonally. During biodegradation experiments, the DOC concentrations of the five sampled water types were found to be significantly different between each other (K-W test, $p < 0.015$; Table S2). In general, the seasonal variability of bioavailable DOC concentrations with incubation time showed a significant increase in values in August and a decrease in October ($p < 0.05$, R$^2$ ranging between 0.32 and 0.85), as well as an insignificant linear increase in BDOC values in May (Figure 4(A1–A5)). This may be due to the initial autumn DOM composition with a lower molecular weight and aromaticity and, accordingly, higher bioavailability. This was in general consistence with the highest initial TBC observed during this season. Within the first 2 days of biodegradation, DOC concentrations slightly decreased (except for the tall ryam and rivers in May), but by no more than 5 mg L$^{-1}$. The corresponding BDOC value increased in incubations or was set to zero (if negative values were obtained, BDOC was taken as 0% following general practice in such experiments, i.e., Vonk et al. [9], Shirokova et al. [15]) from day 0 to day 2 (Figure 4(B1–B5)). After two weeks of incubation, DOC concentrations showed a significant linear decrease (R$^2$ from 0.42 to 0.91, $p < 0.05$) in incubated spring river and autumn-spring mire waters whereas the DOC losses in August were negligible (R$^2$ < 0.25, $p < 0.05$, BDOC = 0–1.8%). Accordingly, DOC reached the maximum biodegradation by the 14th day of incubation in all river waters in May (BDOC = 10.2–18.1%) and in mire waters sampled in spring and autumn (BDOC = 6.5–25.2% and BDOC = 9.9–10.3%, respectively). The loss of DOC was then negligible during the remainder of the incubation in autumn mire waters (R$^2$ from 0.47 to 0.73, $p < 0.05$). In the other cases, after 28 days of incubation, a decrease in BDOC to 0–1.6% was observed; that is, an increase in DOC could occur almost to the initial values. Some negative DOC values at the end of the experiment may be due to the capacity of some microbes to also produce DOC during DOM degradation [57], possibly reducing the net change in DOC concentration. The highest DOC removal rate over 28 days ($R_{DOC}$) was observed in all spring season waters (0.26 ± 0.03 mg L$^{-1}$ day$^{-1}$), while the lowest was usually during October (0.01 ± 0.005 mg L$^{-1}$ day$^{-1}$). The mire waters exhibited the highest $R_{DOC28}$ compared to rivers (Table S3).

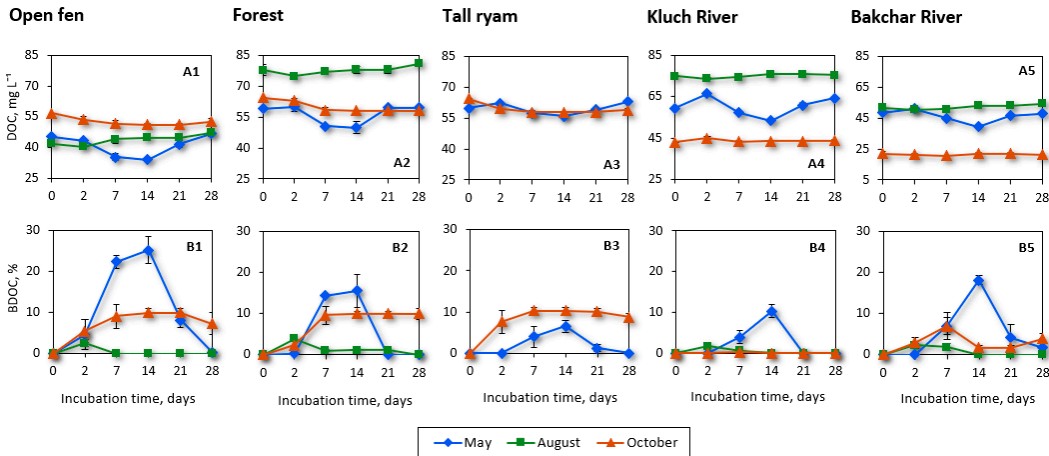

**Figure 4.** DOC concentration (mg L$^{-1}$, (**A1–A5**)) and BDOC (%, (**B1–B5**)) along the aquatic continuum by season during incubation time. The error bars are ±1 SD of the triplicates unless within the symbol size.

Overall, the obtained results on biodegradation at the beginning of the aquatic continuum (fen water and forest water) are comparable to average numbers available from other biodegradation studies as compiled by Vonk et al. [9]: the BDOC fraction over a 28 d exposure period ranges from 3% to 18% in waters of a continuous permafrost zone and from 5% to 15% in a discontinuous permafrost zone. The BDOC values at the end of aquatic continuum (stream and river) are comparable to the values measured for various water bodies including the large river Pechora in the discontinuous permafrost zone of NE European tundra [15]. Our results are also consistent with another study of the aquatic continuum of peatlands, performed in the discontinuous permafrost zone of Western Siberia [30]. These authors reported the biodegradable dissolved OC (BDOC$_{15}$; % DOC lost relative to the initial DOC concentration after 15 days' incubation at 20 °C) as ranged from 0 to 20% for small water bodies located at the beginning of the continuum (soil solutions, small ponds, fens and lakes) and from 10 to 20% for streams and rivers.

The range of DOC biodegradation rates measured in the present study (0.1 to 0.2 mg C L$^{-1}$ d$^{-1}$, Table S3) allows the estimation of the maximal $CO_2$ emissions from water surfaces, assuming the negligible uptake of C for bacterial biomass production, constant biodegradation intensity over the full depth of the photic layer of the water column and postulating that the entire DOC pool is available for degradation. Considering that the photic layer depth in the humic waters of the peatlands does not exceed 0.50 m [15], we calculate that between 0.05 and 0.1 g C-$CO_2$ m$^{-2}$ d$^{-1}$ can be emitted from the surface waters of the considered peatland and river/stream setting during spring flood, but these values decrease significantly in rivers during the other seasons of the year. Even if one considers that 90% of biodegradation occurs over the first two, not four, weeks of incubation (see Figure 4), the resulting potential emissions are still an order of magnitude lower than the actually measured $CO_2$ emissions in the rivers and lakes of Siberian wetlands (1 to 3 g C-$CO_2$ m$^{-2}$ d$^{-1}$, [37,58]). These results corroborate the estimations of Payandi-Rolland et al. [30], who demonstrated that the potential maximum $CO_2$ production from DOC biodegradation in permafrost-bearing surface waters could account for only a small part of the in situ $CO_2$ emissions measured in the peatland aquatic systems of northern Sweden and Western Siberia. Together with these authors, we therefore suggest that other sources (DOC photodegradation, POC biodegradation, respiration of plankton and peryphyton, diffusion from sediments and the lateral influx of $CO_2$-rich soil waters) are likely to be resposnible for the much higher $CO_2$ flux observed in the surface waters of the WSL compared to our back-of-an-envelope calculations based on restricted laboratory incubations. Note that, in addition to DOC biodegradation, the biodegradation of particulate organic matter, which is an order of magnitude more efficient than that of DOM (e.g., [27]), may constitute an important driver of $CO_2$ production in the water column. Furthermore, the photolysis of

DOM and testing the potential link between photochemically reactive soil-derived DOM and stream $CO_2$ fluxes as highlighted elsewhere [32,59,60], have not been performed in Western Siberian Lowland waters and should be a subject for further research.

### 3.3. Optical Characteristics of CDOM during Incubation

The derived CDOM parameters (E2:E3, E2:E4, SUVA$_{254}$, WAMW, S$_{275–295}$, S$_{350–400}$ and S$_R$) showed significantly different values among the five types of water samples (K-W test, $p < 0.00001$; Table S2). The SUVA$_{254}$ ranged between 2.8 and 6 L mg$^{-1}$ m$^{-1}$, showing a seasonal decrease in the order May > August > October, which corresponds to the initial changes in this parameter (Figure 5). Note that such high SUVA values are consistent with the previously reported data of water from the depressions, lakes and rivers of discontinuous permafrost peatlands (e.g., 3.3 to 4.4 L L mg$^{-1}$ m$^{-1}$, [15,29] and peat leachate (6.6 ± 0.1 L mg$^{-1}$ m$^{-1}$, [29]), and likely reflect the dominance of the allochthonous DOM of peat, providing the majority of DOC input to the water bodies.

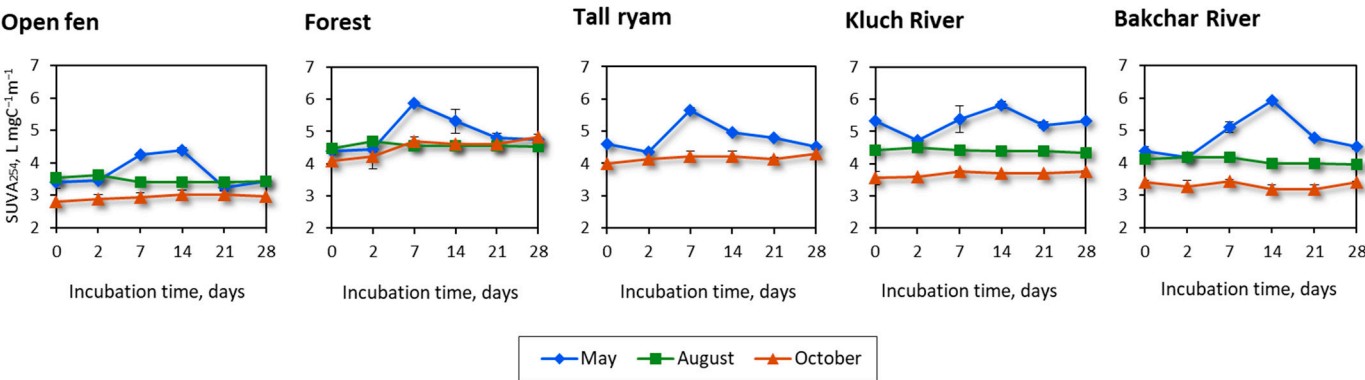

**Figure 5.** SUVA$_{254}$ (L mg C$^{-1}$m$^{-1}$) along the aquatic continuum by season during incubation time. The error bars are ±1 SD of the triplicates unless within the symbol size.

During the incubations, SUVA$_{254}$ values increased in May and October and decreased in August for all sampled water types. The same seasonal and spatial pattern is typical for WAMW, showing positive correlations with SUVA$_{254}$ (r = 0.81–0.92, $p < 0.05$) (Figure S1(E1–E5)). The lowest WAMW values were recorded in autumn (from 1503 to 2290 Da), and the highest in summer (from 2127 to 2742 Da). The linear negative relationship between bioavailable DOC concentrations and SUVA$_{254}$, WAMW (R$^2$ ranged between 0.36 and 0.85, $p < 0.05$) demonstrated that, with an increase in incubation time and%BDOC, the preferential removal of aliphatic low molecular weight DOM occurs in autumn, whereas high molecular weight, aromatic DOM accumulates in spring and summer. These results are consistent with the findings in the permafrost-impacted aquatic continuum, where an increase in the humification of DOM and a decrease in the concentration of aromatic components increased the biodegradable potential of DOM [30].

The slope ratio of the spectrum varied over time from 0.64 to 0.91; the highest values were observed in October and in particular in river waters, indicating the presence of primarily low molecular weight (LMW) material and lower amounts of aromatic functional groups (decreasing SUVA$_{254}$, R$^2$ = 0.52–0.74, $p < 0.05$) (Figure S2(C1–C5)). The presence of these LMW compounds in surface waters during October may be indicative of leaching from litterfall, and this is consistent with elevated R$_{DOC}$ in the forest site during this period (see Table S3). Finally, the E2:E3 and E2:E4 ratios decreasing slightly towards the end of incubation negatively correlated with SUVA$_{254}$ (r = −0.32–0.65, $p < 0.05$) (Figure S1(C1–C5,D1–D5)).

Overall, results of the present study support the former conclusions on DOM evolution along a soil–stream–river continuum in the Canadian boreal zone, which demonstrated that soil–stream waters were a hotspot of DOM degradation, with the elective removal of

LMW components, whereas stream–river waters were more dominated by the degradation of humic-like aromatic components [61].

## 4. Conclusions

A laboratory study of DOM biodegradation from the natural aquatic continuum "mire–forest–stream–river" of the boreal, permafrost-free zone in the Western Siberian Lowland demonstrated the systematic evolution of biodegradable DOC among the sites and across the seasons. The highest biodegradation rates ($R_{DOC}$) were measured in May, during spring flood, when there was a clear decrease in $R_{DOC}$ along the aquatic continuum in the order "open fen ~ forest >> tall ryam $\geq$ stream > river". In summer, this order generally persisted, with higher $R_{DOC}$ in the large river compared to small stream, probably due to the presence of autochonously produced DOM processes in the former. During the autumn, the stagnant surface waters from the forest exhibited maximal biodegradation rates, probably due to the enhanced degradation of the bio-available DOM from the litterfall.

The evolution of the optical parameters of DOM during the incubation showed a general increase in $SUVA_{254}$, likely due to the preferential microbial uptake of aliphatic, optically transparent compounds, as was also consistent with the pattern of the seasonal evolution of other optical parameters among the studied water samples, indicative of DOM molecular weight and aromaticity.

The mass balance calculation of the maximal possible $CO_2$ production from DOM as measured by incubation experiments in the present study yield maximally possible flux in the range of 0.1 and 0.2 g $C\text{-}CO_2$ $m^{-2}$ $d^{-1}$ for the water bodies with a photic layer less than 0.5 m deep. This is an order of magnitude lower than the actual net $CO_2$ emissions from WSL inland waters measured by floating chambers in earlier works. As such, although the biodegradation of humic waters in the fens, forests and rivers of the WSL can sizably modify the concentration and nature of DOM along the aquatic continuum, it plays only a subordinary role in overall C emissions from lakes and rivers of the region.

**Supplementary Materials:** The following supporting information can be downloaded at: https://www.mdpi.com/article/10.3390/w14233969/s1, Table S1: Primary data on chemistry of water during experiments (average of 3 replicates), together with microbial counts; Table S2: Statistical differences in physico-chemical parameters and UV-visible absorbance data between the seasons for each water-sampling location; Table S3: The DOC rate removal at 28 days of incubation ($R_{DOC28}$), *p*-value and $R^2$ coefficient along the aquatic continuum by season on the total period of incubation; Figure S1: Evolution of pH (A), specific conductivity (B), E254:E436 (C), E250:E365 (D) ratios, and WAMW (E) along the aquatic continuum by season during incubation time; Figure S2: Evolution of spectral ratios $S_{275-295}$ (A) and $S_{350-400}$ (B) as well as the slope ratio $S_R$, equaled to $S_{275-295}/S_{350-400}$ (C) along the aquatic continuum by season during incubation time.

**Author Contributions:** Funding acquisition, project administration, conceptualization, investigation, writing—original draft preparation, T.V.R.; formal analysis, visualization S.V.S. and I.V.L.; visualization, investigation, G.I.I.; resources, formal analysis, S.P.K. and E.A.G.; conceptualization, writing—original draft preparation, L.S.S. and O.S.P. All authors have read and agreed to the published version of the manuscript.

**Funding:** The reported study was supported by the Russian Foundation for Basic Research (projects №19-29-05209, №19-35-60030) and the Russian Science Foundation (project №21-77-00021). Laboratory and field research was carried out within the framework of the RSF grant. Data acquisition and manuscript preparation were performed under the RFBR grant.

**Data Availability Statement:** Please note that all the data obtained in this work are available in the main text and in the Supplement of this article.

**Conflicts of Interest:** The authors declare no conflict of interest.

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
