# Peer review of "Seasonal and Spatial Variations of Dissolved Organic Matter Biodegradation along the Aquatic Continuum in the Southern Taiga Bog Complex, Western Siberia"

_water, doi:10.3390/w14233969_

Round 1
Reviewer 1 Report
The authors describe an investigation of the microbial processing of allochthonous organic matter along an aquatic continuum of a wetland watershed. Operationally defined, biodegradable dissolved organic carbon (BDOC) and aqueous optical properties were measured along a mire-forest-stream-river continuum. The manuscript is well written and the results of the incubation experiments to estimate BDOC are consistent with the measurements of optical properties. The paper will advance understanding of seasonal variations of the microbial processing of terrestrial organic matter in a taiga ecosystem and should be published.
A major concern with the conclusions is the estimate of a maximum CO2 flux from the incubation experiments. The authors compare their estimates with measurements of net CO2 emissions from inland waters of the Western Siberian Lowland that were measured by floating chambers and state the flux estimates are an order of magnitude lower. Net gaseous emissions can be directly measured by chamber techniques. The incubation experiments provide information on CO2 production; however, it is highly questionable that they can be used to infer CO2 flux. Seasonal variations of the partial pressure of CO2 in air and water and estimates of overall mass transfer coefficients across the aquatic continuum would be required to apply mass transfer theory to estimate the CO2 flux for the study area. I suggest the estimates of CO2 flux be removed from the manuscript.
A minor editorial suggestion for the Introduction would be to separate the 2 paragraphs that compose the section into several shorter paragraphs to make it easier for the reader to follow the material.
Reviewer 2 Report
In this work, the authors reported the spatial and seasonal variation of dissolved organic matter biodegradation in mire landscape profile located in The Western Siberian Lowlands to reveal its role in the carbon cycle. The content of this article still needs to be added and I recommend its publication in Water after major revision. A number of specific comments are made as follows:
1. Is DOM related to CO2 adsorption in the atmosphere?
2. In introduction, the degradation process of DOM should be further described and discussed.
3. In section 3.1,If it is accidental to take only one sample in each of the three months. Please add the water quality of each season after increasing the number of sampling months and frequency.
4. In the biodegradation experiments, can water samples that are also used only once represent the seasonal variation of BDOM?
5. What is the reason for the difference in DOM degradation between seasons in Figure 4? Please add relevant experiments to prove it.
6. The SUVA section in Figure 4 can be placed in 3.3.
7. Why is August data missing from Figure S1 and Figure S2?
8. The A and B in Figure S2 are mislabeled.
9. The conclusion should be part four and not part five.
Round 2
Reviewer 2 Report
The paper can be accepted in the current form.
Author Response
Dear Reviewer,
We thank you for very constructive remarks that allowed improving our paper. The authors thank you for working on the text of the article and for giving us the opportunity to publish this article.